# Isomelezitose Overproduction by Alginate-Entrapped Recombinant *E. coli* Cells and In Vitro Evaluation of Its Potential Prebiotic Effect

**DOI:** 10.3390/ijms232012682

**Published:** 2022-10-21

**Authors:** Martin Garcia-Gonzalez, Fadia V. Cervantes, Ricardo P. Ipiales, Angeles de la Rubia, Francisco J. Plou, María Fernández-Lobato

**Affiliations:** 1Departamento de Biología Molecular, Centro de Biología Molecular Severo Ochoa (CSIC-UAM), Universidad Autónoma de Madrid, C/Nicolás Cabrera, 1, 28049 Madrid, Spain; 2Instituto de Catálisis y Petroleoquímica, CSIC, 28049 Madrid, Spain; 3Departamento de Ingeniería Química, Universidad Autónoma de Madrid (UAM), Campus de Cantoblanco, 28049 Madrid, Spain

**Keywords:** isomelezitose, immobilized cells, prebiotic agent, *Lactobacillus casei*, *Lactobacillus rhamnosus*, *Enterococcus faecium*, organic acids, α-glucosidase, *Metschnikowia reukaufii*

## Abstract

In this work, the trisaccharide isomelezitose was overproduced from sucrose using a biocatalyst based on immobilized *Escherichia coli* cells harbouring the α-glucosidase from the yeast *Metschnikowia reukaufii*, the best native producer of this sugar described to date. The overall process for isomelezitose production and purification was performed in three simple steps: (i) oligosaccharides synthesis by alginate-entrapped *E. coli*; (ii) elimination of monosaccharides (glucose and fructose) using alginate-entrapped *Komagataella phaffii* cells; and (iii) semi-preparative high performance liquid chromatography under isocratic conditions. As result, approximately 2.15 g of isomelezitose (purity exceeding 95%) was obtained from 15 g of sucrose. The potential prebiotic effect of this sugar on probiotic bacteria (*Lactobacillus casei*, *Lactobacillus rhamnosus* and *Enterococcus faecium*) was analysed using in vitro assays for the first time. The growth of all probiotic bacteria cultures supplemented with isomelezitose was significantly improved and was similar to that of cultures supplemented with a commercial mixture of fructo-oligosaccharides. In addition, when isomelezitose was added to the bacteria cultures, the production of organic acids (mainly butyrate) was significantly promoted. Therefore, these results confirm that isomelezitose is a potential novel prebiotic that could be included in healthier foodstuffs designed for human gastrointestinal balance maintenance.

## 1. Introduction

The human gut microbiota is a complex ecosystem comprised of bacteria, archea, yeast and protozoa that plays an essential role in the maintenance of the host health due to its metabolic activity and physiological interaction with the intestinal mucosa [1,2,3]. Among these microbial symbionts, lactic acid bacteria (LAB), including *Lactobacillus* spp., *Bifidobacterium* spp. and *Entrococcus* spp., are the most representative probiotic bacteria as their proliferation generates benefits in the general health status of the host [4,5]. Thus, several in vitro and in vivo assays demonstrated that the colonization of the human gastrointestinal tract (mainly the colon) by probiotic bacteria enhanced the nutrient and energy absorption, shaped the intestinal epithelium, inhibited the proliferation of pathogenic bacteria and even modulated the immune system [2,6]. Most of the beneficial properties associated with probiotic bacteria derive from the bioactive compounds produced in their metabolic processes (principally fermentation) such as short-chain fatty acids (SCFAs), hydrogen peroxide, diacetyl and bacteriocins [7].

In this scenario, several efforts have been made by the food industry to develop or characterise novel potential prebiotic agents that are selectively fermented by beneficial probiotic bacteria, mainly from the colon, promoting its growth or metabolic activity [8]. It is important to highlight that, based on the current definition of prebiotics [9], a novel ingredient will be considered as a prebiotic agent if its consumption is associated with health benefits to the host, an aspect that would require in vivo assays. The most recognised prebiotic agents are the short carbohydrates fructo-oligosaccharides (FOS), galacto-oligosaccharides (GOS) and isomalto-oligosaccharides (IMOS), although the gluco-oligosaccharides (GlcOS), hetero-glucooligosaccharides (hetero-GlcOS) and cyclo-oligosaccharides have been recently included as potential prebiotic sugars [10,11,12]. The hetero-GlcOS are oligosaccharides comprised of α-glucose monomers linked to at least one non-glucosidic unit as occurs with leucrose, maltulose, isomaltulose, erlose and melezitose [12]. These compounds constitute the minor sugar fraction of honey, a natural sweetener mainly composed of carbohydrates (70–80%) and principally fructose and glucose (~40% and 30% of total carbohydrates, respectively) [13,14]. Water (15–20%) and other minor components (5–10%) such as minerals, phenolic compounds, vitamins or organic acids are also present in this natural product [14]. Honey has been considered as a functional food due to its antimicrobial and prebiotic activity, properties that are directly associated with the aforementioned minor components, including the hetero-GlcOS [15,16]. Thus, different honey oligosaccharide fractions were employed using in vitro assays with faecal bacteria proving that these sugars had a potential prebiotic effect, but not to the level that FOS had [17]. Later, a study performed with static faecal batch cultures demonstrated that melezitose, a recognised hetero-GlcOS from honey, was fermented more slowly than other oligosaccharides such as sucrose, raffinose and lactulose [18]. Therefore, this sugar would have the potential to reach the distal colon intact and be fermented by the beneficial bacteria located there.

The in vitro evaluation of the potential prebiotic activity of complex sugars usually requires their removal from other carbohydrates also contained in the mixture, especially the majority ones and very frequently glucose and fructose [19]. The partial purification of specific sugars in these complex mixtures could be performed by physical methods using activated charcoal and nanofiltration, or by biotechnological approaches employing enzymes or whole microorganisms [17,20]. Thus, *Komagataella phaffii* cells immobilized in calcium alginate beads have recently been used for the elimination of glucose and fructose from different sugar mixtures [21].

In previous works, our group expressed in *Escherichia coli* and characterized an α-glucosidase (Mr-αGlu) responsible for a glucosyltransferase activity detected in cell extracts from the nectar yeast *Metschnikowia reukaufii* [12,22]. The enzyme Mr-αGlu produced a mixture of honey hetero-GlcOS by transglucosylation of sucrose, with isomelezitose as the principal product [12]. Isomelezitose is mainly synthesised by microbial α-glucosidases included in the family GH13 [12,23,24]. Thus, the production of this trisaccharide from sucrose (100 g/L) was firstly described using the *Saccharomyces cerevisiae* α-glucosidase, where isomelezitose yield was lower than 0.5% of the total sugar in the reaction mixture [25]. Better results were obtained employing the α-glucosidase from *Bacillus* spp. and the sucrose isomerase from *Protaminobacter rubrum*; both enzymes improved by protein engineering, obtaining approximately 3% and 22% of isomelezitose, starting from 600 and 100 g/L sucrose, respectively [23,26]. More recently, the transglucosylation activity of the maltase from the yeast *Blastobotrys adeninivorans* was analysed using 513 g/L sucrose [27]. In this work, the maximum isomelezitose amount was about 25% of total sugars, an enhanced result, but still lower than the yield reported by our group using Mr-αGlu (approximately 35% of total sugars) [12].

Isomelezitose is a rare trisaccharide commonly found in plant-associated materials, in particular honey, and therefore suitable for human consumption. In addition, isomelezitose has a great biotechnological relevance, for instance, it could be used as a sugar substitute in different foodstuff due to its non-cariogenic effect and low calorific value in comparison with dextrose; and as a potential osmoprotectant of microbial cells during freeze-drying processes [23,28]. Additionally, isomelezitose has been proposed as a prebiotic candidate [24], but as far as we know, this bioactivity has not been analysed yet. In this study, *E. coli* cells expressing the α-glucosidase from *M. reukaufii* were entrapped in alginate beads and used to overproduce isomelezitose from sucrose. Monosaccharides from the resulting oligosaccharide mixture were removed using *K. phaffii* cells, also immobilized in alginate beads. Then, isomelezitose was efficiently purified by semi-preparative chromatographic techniques. Finally, the potential prebiotic activity of this trisaccharide was evaluated for the first time using probiotic bacteria of the genera *Lactobacillus* and *Enterococcus*.

## 2. Results and Discussion

### 2.1. Production and Characterisation of a Biocatalyst Based on Alginate-Entrapped E. coli Cells

Since the α-glucosidase from *M. reukaufii* (Mr-αGlu) had already been successfully expressed in *E. coli* without apparent changes in its biosynthetic activity [12], we decided to prepare a biocatalyst based on alginate-entrapped *E. coli* cells expressing this protein. Maximum Mr-αGlu activity was obtained at the mid-log phase of the bacteria culture, 1.0 mM IPTG and 16 h of induction (Figure 1a, b and c, respectively). Additionally, different amounts of bacterial cells were also entrapped in calcium alginate beads as referred in the Material and Methods Section. The best results of the enzyme’s apparent hydrolytic activity were obtained with a cell concentration of 15% (*w*/*v*), with no significant improvements produced by increasing this concentration (Figure 1c). 

The hydrolytic activity of the immobilized-cells biocatalyst expressing Mr-αGlu was evaluated after 24 h incubation at different temperature values. It was stable in the range of 4–30 °C and retained the maximal residual activity at 30 °C with an insignificant hydrolytic activity detected at temperatures ≥35 °C. The non-immobilized protein Mr-αGlu (Figure 2a) showed a similar thermal behaviour compared to the alginate-entrapped biocatalyst.

The capacity of the immobilized *E. coli* cells to produce honey hetero-GlcOS by transglucosylation of sucrose was evaluated using high performance anion-exchange chromatography with pulsed amperometric detection (HPAEC–PAD) analysis. A representative chromatographic profile of the reactions performed with non-induced and induced *E. coli* cells containing the construction MrαGlu-pET28b(+) is shown in Appendix A. All chromatographic signals obtained in these experiments corresponded to the peaks previously detected using the soluble Mr-αGlu purified from the *E. coli* cultures [10]. Thus, the trisaccharide isomelezitose [α-D-Glc-(1→6)-β-D-Fru-(2→1)-α-D-Glc] and the disaccharide trehalulose [α-D-Glc-(1→ 1)-β-D-Fru] were the main transglucosylation products, followed by trisaccharides such as erlose [α-D-Glc-(1→4)-α-D-Glc-(1→2)-β-D-Fru], melezitose [α-D-Glc-(1→3)-β-D-Fru-(2→1)-α-D-Glc] and theanderose [α-D-Glc-(1→ 6)-α-D-Glc-(1→2)-β-D-Fru], which were synthesised as minor compounds. The isomelezitose production by the entrapped *E. coli* cell biocatalyst was followed for 54 h by HPLC. The maximum concentration of isomelezitose was reached after 24 h of reaction (~62 g/L) and remained constant for at least 30 more hours (Figure 2b). These results confirmed that alginate-entrapped *E. coli* cells harbouring the α-glucosidase Mr-αGlu is a suitable biocatalyst for the hetero-GlcOS production. However, with the aim of developing a Generally Recognized as Safe (GRAS) process for honey oligosaccharides synthesis (in particular isomelezitose), other heterologous organisms overexpressing the Mr-αGlu activity, such as *S. cerevisiae* or *K. phaffii*, could substitute *E. coli* cells.

The reusability of the immobilized *E. coli* cells was also evaluated by successive transglucosylation reactions in batches (26–28 h each cycle). Compared to the first reaction cycle, in which isomelezitose concentration reached ~67 g/L, the immobilized biocatalyst produced about 90% of this initial amount after 5 reaction cycles (~135 h), and approximately 65% in the cycle 6 (~160 h; Figure 2c). These operational stability results led us to the application of the immobilized biocatalyst in large-scale production of potential prebiotic honey hetero-GlcOS, in particular the trisaccharide isomelezitose.

### 2.2. Isomelezitose Overproduction Employing the Immobilised E. coli Cells in Alginate Beads

The hetero-GlcOS were overproduced in batch reactions using alginate-entrapped *E. coli* [MrαGlu-pET28b(+)] cells according to the procedure described in Section 3.7, and the sugar composition of the reaction mixture is indicated in Table 1. Although the process for obtaining these sugars was scaled up by increasing the reaction volume from 1 to 25–30 mL, and considerably simplified since no Mr-αGlu purification step was required, the total yield of hetero-GlcOS produced was reduced by about 15%, and mainly due to the production of isomelezitose that was reduced by more than 50%.

In addition, the maximum concentration of trehalulose as well as the rest of the total hetero-GlcOS practically doubles the amounts produced by the soluble enzyme (Table 1). There was a noticeable increase in the production of a non-identified disaccharide (31 g/L) in the reactions catalysed by alginate-entrapped *E. coli* cells (Appendix A). Considering the results reported for the overload sucrose reactions using the α-glucosidase from the yeast *B. adeninivorans*, this unknown compound could correspond to the possible disaccharides: maltulose [α-D-Glc-(1→4)-β-D-Fru], turanose [α-D-Glc-(1→3)-β-D-Fru] or isomaltulose [α-D-Glc-(1→6)-β-D-Fru] [27]. The aforementioned results pointed to an apparent increase in the hydrolytic processes that could promote a major free fructose glucosylation. This behaviour was previously described for the β-fructofuranosidase from the yeast *Xanthophyllomyces dendrorhous* entrapped in polyvinyl alcohol lenses, where the hydrophilic microenvironment generated inside the carrier could have increased the hydrolysis/transglycosylation ratio [29]. In this context, different authors have already reported the employment of alginate-entrapped whole cells for bioactive oligosaccharides production. Thus, approximately 265 g/L of FOS were obtained from 600 g/L sucrose and 240 g/L of GOS from 600 g/L lactose using immobilised *Aureobasidium pullulans* and *Sporobolomyces singularis* cells, respectively [30,31].

The sugar mixture produced by the immobilised *E. coli* cells expressing Mr-αGlu was enriched in hetero-GlcOS by adding alginate-entrapped *Komagataella phaffii* cells, a very efficient and selective cellular system recently employed to remove glucose and fructose in carbohydrate syrups [21]. Thus, after the reaction mixture was incubated with the immobilised yeast cells (~30 h), the monosaccharides were reduced from the initial 43.5% to 1.8% (*w*/*w*) referred to the total sugars of the mixture. Consequently, the mixture was enriched in di- and trisaccharides, which increased from the initial 56.5% to 98.2% (*w*/*w*) of the total sugars (Appendix A). 

### 2.3. Potential Prebiotic Effect of the Hetero-GlcOS Generated by the Alginate-Entrapped E. coli Cells Expressing Mr-αGlu

Initially, the potential prebiotic activity of the honey hetero-GlcOS mixture produced by the immobilised biocatalyst was evaluated by an in vitro micro-scale assay using three probiotic bacterial strains: *Lactobacillus casei*, *Lactobacillus rhamnosus* and *Enterococcus faecium*. As controls, microbial cultures were prepared in parallel with no carbon source (negative control 1) and supplemented with the commercial FOS mixture Actilight (positive control), mainly composed by 1-kestose and nystose (Appendix A), two sugars widely recognized as prebiotic agents [32]. In addition, microbial cultures based on Man-Rogosa-Sharpe (MRS) medium supplemented with a mixture of fructose, glucose and sucrose in the same concentrations as found in the hetero-GlcOS mixture (0.36, 0.18 and 6.9% *w*/*v*, respectively) were also prepared and used as negative control 2. All probiotic bacteria analysed in this work grew in MRS cultures modified with the honey oligosaccharides mixture obtained with the entrapped *E. coli* cells harbouring the Mr-αGlu activity, as evidenced by the growth curves shown in Appendix A. As expected, in cultures where no carbon source was added, the probiotic bacteria growth was very limited. Maximum values of turbidity (*Δ_max_*OD_600_) after 24 h incubation were reached for *L. rhamnosus* and *E. faecium* when the hetero-GlcOS mixture were used as carbon source, and for *L. casei* when it was supplemented with FOS (Figure 3). In addition, FOS enhanced the growth of *L. rhamnosus*, but not of *E. faecium* (*p* < 0.05). These results are in accordance with previous reports where honey oligosaccharides showed a positive influence in the growth of *Lactobacillus* and *Bifidobacterium* spp. [17,33].

### 2.4. Analysis of the Potential Prebiotic Activity of Isomelezitose

To evaluate the potential prebiotic activity of isomelezitose (the main honey trisaccharide produced by Mr-αGlu), this compound was purified from the sugar mixture previously enriched in hetero-GOS by treatment with immobilised *K. phaffii* cells using HPLC in a semi-preparative scale. Thus, the recovered isomelezitose, approx. 2.15 g from 15 g of sucrose (90–95% of recovery) was obtained with a purity higher than 95% (*w*/*w*) (Figure 4). It should be highlighted that in the sugar mixture treated with *K. phaffii*, new peaks with lower retention times were detected by HPAEC–PAD analysis (Figure 4). These unknown peaks might correspond to oligosaccharides with mannose residues or *N*-acetylglucosamine liberated from the yeast cell wall, or even organic compounds produced by the yeast cellular metabolism.

Since chromatographic separation of oligosaccharides from complex sugars mixtures is expensive and difficult to scale-up, nanofiltration stands out as an easy and cost-effective methodology used for mono- and oligosaccharides fractionation [34]. Thus, oligosaccharides were effectively removed (70% yield) from monosaccharides and lactose in a GOS mixture using a membrane with a molecular weight cut off (MWCO) of 800–1000 Da [35]. In addition, xylo-oligosaccharides and xylan were purified from a mixture that contained xylose and other by-products through nanofiltration with discontinuous diafiltration, obtaining oligosaccharides recovery of approximately 84% [36]. Therefore, nanofiltration could be an interesting alternative to purify the hetero-GlcOS mixture produced by Mr-αGlu activity at pilot- or industrial-scale. 

The potential prebiotic effect of isomelezitose was tested on the aforementioned probiotic bacteria using the in vitro micro-scale assay previously described. Negative and positive controls were also employed in these assays and results are shown in Table 2. The maximal cell densities reached in cultures supplemented with isomelezitose were significantly higher than the negative controls, and similar to the bacteria inoculated in MRS medium with the commercial FOS mixture. Concerning the generation time (*g*), all probiotic bacteria displayed similar specific growth rates in isomelezitose-based media compared to the positive controls and as expected, these values were significantly higher than the negative controls. Therefore, isomelezitose could be selectively used as a carbon source by the beneficial probiotic bacteria promoting its growth, in a similar way that FOS do. These results are in accordance with a previous research work where a mixture of partially purified honey oligosaccharides promoted the growth of *Lactobacillus* and *Bifidobacteria*, but not in the same level as FOS did [17]. In addition, the ingestion of isomaltulose [α-D-Glc-(1→6)-β-D-Fru], a disaccharide that could be classified also as hetero-GlcOS, had a positive effect in the populations of beneficial faecal bacteria in rats, determined through 16S rRNA sequencing [37].

Figure 5 shows the progress of the short-chain fatty acids (SCFAs: acetate, propionate and butyrate) and lactate concentrations in the different MRS cultures where the probiotic bacteria were inoculated. In all cultures analysed, acetic and lactic acids were the main fermentation end products, as expected for the lactic acid bacteria (LAB). Throughout the incubation of *L. casei* for 24–48 h, the production of acetic and butyric acids was significantly higher in the media supplemented with isomelezitose compared to the negative and positive controls. Concerning the lactate synthesis, *L. casei* grown in the presence of FOS and isomelezitose produced similar amounts of this organic acid that were statistically higher than the lactate production in MRS with no carbon source. *L. rhamnosus* displayed a similar behaviour concerning the organic acids production compared to *L. casei*, with acetate and lactate as the main products. Again, butyrate concentration was significantly higher in MRS media with isomelezitose compared to the media without carbon source or supplemented with the commercial FOS mixture. Finally, for *E. faecium* cultures, no significant differences were observed in the concentrations of the organic acids in the media modified with isomelezitose or FOS and the negative controls, at 24 h of incubation. However, at 48 h of incubation, the concentrations of acetate, butyrate and lactate in isomelezitose-based cultures were significantly higher than the MRS media with no carbon source and the media supplemented with the FOS mixture. For all the strains tested and under all the conditions employed, no significant differences were detected for propionate concentrations. The major production of acetate in comparison with lactate was previously reported for *L. casei* and *L. plantarum* in MRS media, probably because of the heterofermentative pathway where acetic acid is the principal product [7]. Organic acids (including SCFAs and lactate) produced by LAB are considered as important metabolites for the maintenance of a balanced human gut microbiota, principally due to the pH reduction effect that inhibits the pathogens’ proliferation [38]. In addition, SCFAs produced during fermentation of prebiotics, in particular dietary fibre and non-digestible oligosaccharides, are associated with a reduction in the risk of suffering cardiovascular diseases, type 2 diabetes and even inflammation and potential mental disorders [1,6,39,40]. In this work, it is remarkable that the addition of isomelezitose to the probiotic bacteria cultures promoted a considerable higher butyrate production compared to the media with no carbon source, and even to the cultures supplemented with the same amount of FOS. Butyrate is the main energy source for colonocytes maintaining the gut epithelium integrity and some studies have demonstrated that this SCFA induces apoptosis in colorectal tumour cells [40,41].

## 3. Materials and Methods

### 3.1. Materials and Reagents

Glucose, fructose, and sucrose were from Merck (Darmstadt, Germany). Melezitose, kanamycin sulphate, isopropyl-β-D-1-thiogalactopyranoside (IPTG), 3,5-dinitrosalicylic acid (DNS) and beef extract were from Sigma-Aldrich (Saint Lois, MO, USA). Polysorbate 80 was from Santa Cruz Biotechnology (Dallas, TX, USA) and 3-(N-morpholino)-propanesulfonic acid (MOPS) from NzyTech (Lisbon, Portugal). Sodium alginate Protanal LF 120 LS was from FMC Biopolymer (Grivan, Ayrshire, UK). Beghin Meiji SA (Neuilly-sur-Seine, France) kindly supplied the commercial fructo-oligosaccharides mixture “Actilight”. Yeast extract, peptone and tryptone were from Laboratorios Conda S.A. (Madrid, Spain).

### 3.2. Microorganisms and Culture Conditions

*Escherichia coli* BL21 (DE3) (Invitrogen; Carlsbad, CA, USA) transformed with the construction MrαGlu-pET28b(+) that contained the gene responsible for the α-glucosidase activity from *Metschnikowia reukaufii* (*Mr-αGlu*) was previously reported [12]. *Komagataella phaffii* GS115 was from Invitrogen (Carlsbad, CA, USA) and the probiotic bacteria *Lactobacillus casei* CECT475, *Lactobacillus rhamnosus* CECT278 and *Enterococcus faecium* CECT410, were all from the Spanish Type Culture Collection (Valencia, Spain).

Expression of the α-glucosidase Mr-αGlu in *E. coli* [MrαGlu-pET28b(+)] cultures was controlled by IPTG and was carried out as described previously [10], but using the optimal conditions determined in this work. The yeast *K. phaffii* was cultured at 30 °C with orbital shaking (250 rpm) in 10 mL of YEPD medium (10 g/L yeast extract 20 g/L peptone and 20 g/L glucose) supplemented with 25 μg/mL chloramphenicol. The growth was monitored spectrophotometrically at 660 nm (OD_660 nm_). After an incubation of 16–18 h, 5 mL of culture was added to a 500 mL flask containing 50 mL of YEPD medium (also with 25 μg/mL chloramphenicol) and left for 16 h at 30 °C and orbital shaking (250 rpm). For both *E. coli* and *K. phaffii* cultures, cells were harvested by centrifugation (5000× *g* for 10 min) and washed three times with PBS 1X (8.0 g/L sodium chloride, 0.2 g/L potassium chloride, 1.4 g/L sodium phosphate dibasic and 0.24 g/L potassium phosphate monobasic) to remove the remaining medium.

*L. casei*, *L. rhamnosus* and *E. faecium* strains were cultured in MRS medium (10 g/L peptone, 10 g/L beef extract, 5.0 g/L yeast extract, 1.0 g/L polysorbate 80, 2.0 g/L ammonium citrate, 5.0 g/L sodium acetate, 0.1 g/L magnesium sulphate, 0.05 g/L manganese sulphate, 2.0 g/L dipotassium sulphate and 2.0 g/L glucose). Before inoculation, the pH medium was adjusted to 6.2–6.4 range with HCl and then autoclaved at 121 °C for 15 min. When necessary, MRS was prepared without glucose or supplemented with 0.5–2.0% (*w*/*v*) of the required carbon source (FOS mixture, hetero-GlcOS or purified isomelezitose). These bacteria were cultured at 37 °C in aerobic conditions.

### 3.3. Optimal Conditions for Mr-αGlu Expression in E. coli

The best conditions for the expression of the α-glucosidase Mr-αGlu in *E. coli* [MrαGlu-pET28b(+)] were analysed using different IPTG concentrations and induction times. Initially, cultures were induced with 1 mM IPTG at different bacteria culture grow phases (OD_600 nm_ ~0.2–1.0) and then kept for 16–18 h with orbital shaking (200 rpm) at 16 °C. Optimal concentration of inducer was assessed by adding to *E. coli* cultures (OD_600 nm_ ~0.8) different amounts of IPTG (final concentration 0.2, 0.5 and 1.0 mM) and incubated overnight as previously described. To analyse the best induction time, induced bacteria (OD_600 nm_ ~0.8 and 1 mM IPTG) were incubated at 16 °C and 200 rpm for different times in the 2–24 h interval. In all cases, cells were harvest by centrifugation (5000× *g* for 10 min), washed three times with phosphate buffer saline (PBS) 1X to remove the remaining medium and suspended in PBS 1X to a final concentration of about 50% (*w*/*v*). Hydrolytic activity of the whole cells suspensions was determined as described in Section 3.4.

### 3.4. Standard Hydrolytic Activity Assay

The hydrolytic activity of immobilised *E. coli* [MrαGlu-pET28b(+)] cells was determined by DNS assay adapted to 96-well plates according to the procedure described elsewhere employing micro-centrifuge filter tubes with 0.45 μm cellulose membrane [42]. For free whole cells, the hydrolytic activity was measured employing cells suspensions in PBS 1X. Reactions were performed at 30 °C for 20 min using 100 g/L sucrose in 50 mM 3-(*N*-morpholino)propanesulfonic acid (MOPS) buffer pH 7.0. A glucose calibration curve in the 0–3 mg/mL range was employed. One unit (U) of hydrolytic activity was defined as that corresponding to the release of one μmol of reducing sugar per minute.

### 3.5. Entrapment of E. coli Cells Harbouring the Mr-αGlu Activity into Alginate Beads

Whole cell immobilisation in alginate beads was carried out by preparing a 4.0% (*w*/*v*) sodium–alginate–aqueous solution stirred for ~2–4 h until a homogeneous clear solution was obtained and no air bubbles were observed. Then, the alginate solution was gently mixed in a ratio 1:1 (*w*/*w*) with the cell suspension of *E. coli* or *K. phaffii* in distilled water (at 30 and 50% *w*/*v*, respectively). The mixture was added by controlled dripping (1 mL/min) using a peristaltic pump (P-1, GE Healthcare) into a 200 mM CaCl_2_ solution and then, the gel beads obtained by ionotropic gelation were maintained under magnetic stirring at 150 rpm. After 10 min, the alginate beads were transferred to a fresh 200 mM CaCl_2_ solution and kept for another 10 min with mild agitation. Finally, the alginate beads were separated from the solution and thoroughly washed with distilled water.

The amount of induced *E. coli* [MrαGlu-pET28b(+)] cells was optimized through the preparation of immobilised biocatalyst with several concentrations of this microorganism (~5.0–25.0%; *w*/*v*) and then, the apparent activity was measured as previously indicated.

### 3.6. Thermal and pH Stability of the Alginate-Entrapped E. coli Cells

The pH and thermal stability of immobilised *E. coli* [MrαGlu-pET28b(+)] cells in calcium alginate beads was determined by pre-incubating ~35 mg (wet weight) of the beads (~1 U) for 24 h at different pH (4.0–9.0) and temperatures (4–40 °C). Afterwards, the apparent hydrolytic activity was determined by the DNS assay using centrifuge filter tubes with 0.45 μm cellulose membrane at 30 °C and employing 100 g/L sucrose. The buffers used (50 mM) were citric acid/sodium citrate (pH 4.0–6.0), MOPS (pH 6–8) and glycine/NaOH (pH 8.0–9.0). All analysis were performed in triplicate.

### 3.7. Production of Honey Hetero-GlcOS and Operational Stability of Immobilised Biocatalysts in Transglucosylation Reactions

Honey hetero-GlcOS were produced by sucrose transglucosylation under the optimal conditions previously reported (500 g/L sucrose, 30 °C and pH 7.0) for the α-glucosidase from *M. reukaufii* [10]. Approximately 7 g (wet weight, ~240 U) of induced alginate-entrapped *E. coli* [MrαGlu-pET28b(+)] cells were incubated with 25–30 mL of sucrose solution and kept for 24–28 h, the time required for reaching a constant bioconversion. Immobilised biocatalyst with non-induced *E. coli* cells were also used as a control in these assays. The analysis was performed in micro-centrifuge filters tubes with 0.45 μm cellulose acetate membranes (Costar Spin-X, Corning Inc.; Corning, NY, USA). Samples were withdrawn at different reaction times and analysed by HPLC-ELSD as referred previously [20].

The reusability of the alginate-entrapped *E. coli* cells was evaluated through several enzymatic cycles. Each cycle was performed at the optimal conditions, using 40–45 mg of alginate beads (about 1.5 U) and 500 μL of a 500 g/L sucrose solution in a micro-centrifuge filter tube. After each cycle, the reaction mixture was separated by centrifugation at 5000× *g* and stored at −20 °C until high-performance anion-exchange chromatography with pulsed amperometric detection (HPAEC–PAD) analysis. Then the beads were washed three times with distilled water to remove the remaining sugars and were used for the next batch reaction. All the reactions were performed in triplicate.

### 3.8. Partial Purification of Hetero-GlcOS Mixture by Alginate-Entrapped K. phaffii and Isomelezitose Purification

Monosaccharides (glucose and fructose) elimination from the hetero-GlcOS mixture was carried out using alginate-immobilised *K. phaffii* according to the procedure described elsewhere [19]. Briefly, the reaction mixture (25–30 mL) obtained with alginate-entrapped *E. coli* was filtered and then incubated at 30 °C with 10 g of *K. phaffii* cells also immobilised in calcium alginate. At different times, aliquots were withdrawn and analysed by HPAEC–PAD until glucose and fructose disappeared. After the monosaccharides were eliminated, the hetero-GlcOS mixture was subjected to an additional purification step by semi-preparative HPLC (Section 3.10).

### 3.9. Effect of Isomelezitose and Commercial Prebiotic Sugars on the Grow and Metabolic Activity of Probiotic Bacteria

The in vitro analysis of the potential prebiotic effect of hetero-GlcOS and isomelezitose were carried out based on a micro-scale assay described elsewhere [43]. In a first step, the growth of commercial probiotic bacteria was evaluated using honey hetero-GlcOS mixture as carbon source in sterile 96-well microplates. MRS medium without carbon source (negative control 1); supplemented with 2.0% (*w*/*v*) commercial FOS (Actilight, positive control) and 2.0% (*w*/*v*) of partially purified hetero-GlcOS mixture were prepared. Additionally, media supplemented with the amount of fructose, glucose and sucrose determined by HPAEC–PAD in the hetero-GlcOS mixture was also prepared and used as negative control 2. All the MRS media were inoculated at 2.0% (*v*/*v*) with overnight growth cultures of the probiotic strains, and then, approximately 200 μL of each media were transferred to 96-well plates (in triplicate) and sealed with gas permeable sealing membranes (Sigma-Aldrich; Saint Lois, MO, USA) under sterile conditions. The microplates were incubated at 37 °C under orbital agitation in a Tecan (Männedorf, Switzerland) plate reader. Turbidity was monitored at 600 nm every 30 min for 24 h incubation. After confirming that probiotic bacteria were able to ferment the hetero-GlcOS mixture, the same micro-scale assay was performed using the purified isomelezitose at 0.5% (*w*/*v*) as carbon source. The same negative and positive controls were prepared using MRS media without carbon source or with the commercial FOS mixture Actilight at 0.5% (*w*/*v*), respectively. Again, the OD_600 nm_ was monitored every 30 min for 24 h and growth kinetic parameters were determined by standard methods, where specific growth rates were defined as Ln2/*g*. The maximum culture grow (*Δ_max_*OD_600_) was calculated according to Equation (1).
*Δ_max_*OD_600_ = OD_600 (24 h)_ − OD_600 (initial)_(1)

In all cases, sterility controls were made by using the MRS media without bacteria.

The metabolic activity of each probiotic bacteria in MRS media supplemented with isomelezitose was assessed by the evaluation of short-chain fatty acids (SCFAs) and lactate formation. Thus, 5 mL of each culture based on MRS media [without carbon source, with 0.5% (*w*/*v*) FOS and with 0.5% (*w*/*v*) isomelezitose] were inoculated with 2% (*v*/*v*) of each probiotic bacteria pre-culture and incubated at 37 °C under aerobic conditions. At 24 and 48 h, 500 μL of each culture was withdrawn and centrifuged at maximum speed (11,740× *g*, Eppendorf MiniSpin) to remove the cells. The supernatants were stored at −20 °C until gas chromatography (GC) or ionic chromatography analysis.

### 3.10. Analytical Analysis

The concentration of sugars in the supernatant was determined by HPAEC–PAD on a Dionex ICS3000 system with a CarboPack PA-1 column (4 × 250 mm) connected to a PA-1 guard column and an electrochemical detector with a gold working electrode and Ag/AgCl reference electrode. The flow rate was fixed at 1 mL/min. The initial mobile phase was 100 mM NaOH for 8 min. Then, a gradient from 0–72 mM sodium acetate maintaining 100 mM NaOH was performed in 22 min. This last mobile phase composition was kept for 6 min and then changed to 100 mM NaOH containing 300 mM sodium acetate. Eluents were degassed by flushing with helium, and peaks were analysed using Chromeleon software. Glucose, fructose, isomelezitose, melezitose monohydrate and sucrose were used as standards.

Isomelezitose formed during the transglucosylation reaction was purified by semipreparative HPLC using a system equipped with a quaternary pump (Delta 600; Waters) coupled to a Luna-NH_2_ column (250 × 10 mm, 10 µm) (Phenomenex; Torrance, CA, USA). A three-way flow splitter (Accurate, LC Packings) and the ELSD detector equilibrated at 90 °C was used. An isocratic method using acetonitrile/water 85:15 (*v:v*) was employed as mobile phase at 5.5 mL/min for 60 min. The column temperature was kept at 30 °C. After collecting the isomelezitose peak, the mobile phase was eliminated by rotary evaporation in an R-210 rotavapor (Buchi, Essen, Germany). The purity of the compounds collected in the semi-preparative HPLC was analysed by HPAEC–PAD using the methods described above.

SCFAs concentrations (from acetic to heptanoic, including iso-forms) were determined by GC (Varian 430-GC) equipped with a flame ionization detector (FID) and a capillary column filled with Nukol (polyethylene glycol modified by nitroterephthalic acid) [44]. Prior to injection, 900 μL of the sample was mixed with 150 μL of phosphoric acid (1:2 *v*:*v*) to adjust pH below 2.0 and 150 μL of a solution of crotonic acid (2000 mg/L) as an internal standard. This mixture was centrifuged to remove any solids and transferred to a 1500 μL GC vial. The sample injection volume was 1 μL. The temperatures of the injector and detector were maintained at 200 and 250 °C, respectively, while the column temperature was increased from 120 to 150 °C with an increasing rate of 10 °C/min.

Lactic acid was quantified by means of ionic chromatography with chemical suppression (Metrohm 790 IC) using a conductivity detector. A 3.2 mM Na_2_CO_3_ aqueous solution was used as mobile phase (flow rate of 0.7 mL/min), 100 mM H_2_SO_4_ was used as suppressor and a Metrosep A sup 5–250 column (250 mm x 4 mm) as stationary phase. All experiments were performed in duplicate and average values were calculated.

### 3.11. Statistical Analysis

Data are presented as the mean values obtained in triplicated experiments with their standard deviations (mean ± SD). One-way ANOVA coupled with Tukey’s test with a 5% confidence interval was used to perform the multiple comparison and to evaluate the significant differences between groups, employing GraphPad Prism (version 6; GraphPad Software, San Diego, CA, USA).

## 4. Conclusions

A mixture of honey hetero-glucooligosaccharides (hetero-GlcOS) were synthesised from sucrose using immobilised *E. coli* cells expressing the enzyme Mr-αGlu. Compared to the soluble enzyme, the immobilised biocatalysts displayed a higher residual activity after 24 h incubation at 30 °C (60% vs. 100%, respectively) and were successfully reused in five transglucosylation cycles of 24 h with no significant loss of activity. High purity isomelezitose (above 95%) was produced in a three-step process. In vitro analysis of the prebiotic activity showed that isomelezitose was beneficial for the growth of probiotic bacteria of the genus *Lactobacillus* and *Enterococcus*. Furthermore, the addition of this trisaccharide to cultures of the aforementioned bacteria promoted the overproduction of important metabolites for human gastrointestinal health such as butyrate, acetate and lactate. Thus, these results represent an excellent approach to a bioprocess focused on the overproduction of potential high-value novel prebiotic agents such as hetero-GlcOS, especially the rare trisaccharide isomelezitose.

## Figures and Tables

**Figure 1 ijms-23-12682-f001:**
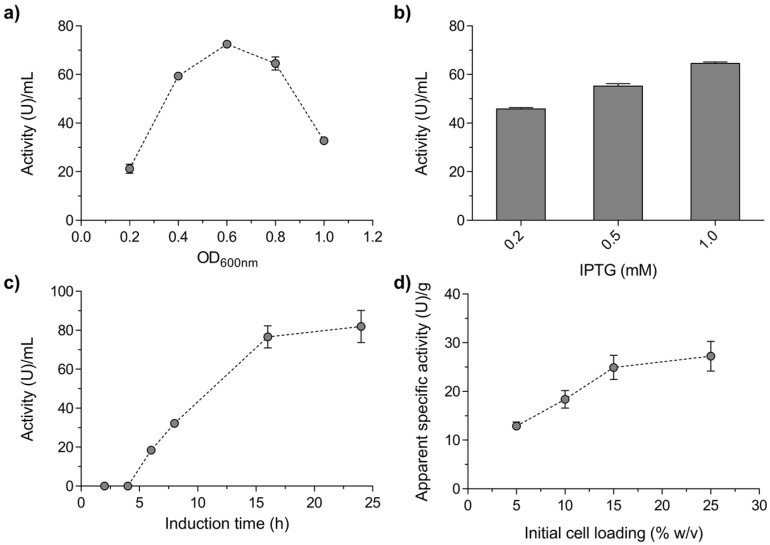
Optimization of Mr-αGlu activity in *E. coli* and alginate beads preparation. The maximum hydrolytic activity of the bacteria cells expressing the protein Mr-αGlu was determined at different: (**a**) optical cell densities (OD600 units) for 16 h induction with 1 mM IPTG, (**b**) IPTG concentration added to *E. coli* cultures (~0.6 OD600 units) incubated for 16 h and (**c**) induction times of *E. coli* cultures (1 mM IPTG and ~0.6 OD600 units). (**d**) Evaluation of the biocatalyst activity in relation to the cells concentration used in the alginate beads preparation. All bacteria cultures were incubated at 16 °C. Hydrolytic activity was determined using 100 g/L sucrose (30 °C, 20 min) by the 3,5-dinitrosalicylic acid (DNS) method. Error bars represent the standard deviations from three *independent* analysis.

**Figure 2 ijms-23-12682-f002:**
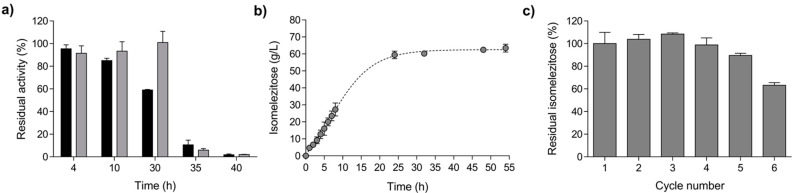
Characterisation of alginate-entrapped *E. coli* [MrαGlu-pET28b(+)] biocatalyst. (**a**) Hydrolytic activity of soluble (black bars) and immobilized (grey bars) biocatalyst after 24 h incubation at different temperatures (pH 7.0) are shown. (**b**) Isomelezitose production using immobilized *E. coli* [MrαGlu-pET28b(+)] cells from 500 g/L sucrose. Reactions were followed for 54 h and different times aliquots were withdrawn and analysed by high performance anion-exchange chromatography with pulsed amperometric detection (HPAEC–PAD). (**c**) Operational stability of immobilized biocatalyst in transglucosylation reaction cycles of 24–26 h each. Reaction conditions: ~43 mg of alginate-entrapped cells (~2 U), 500 g/L sucrose and 30 °C. In all cases, the results shown corresponded to three independent measurements with the standard deviation.

**Figure 3 ijms-23-12682-f003:**
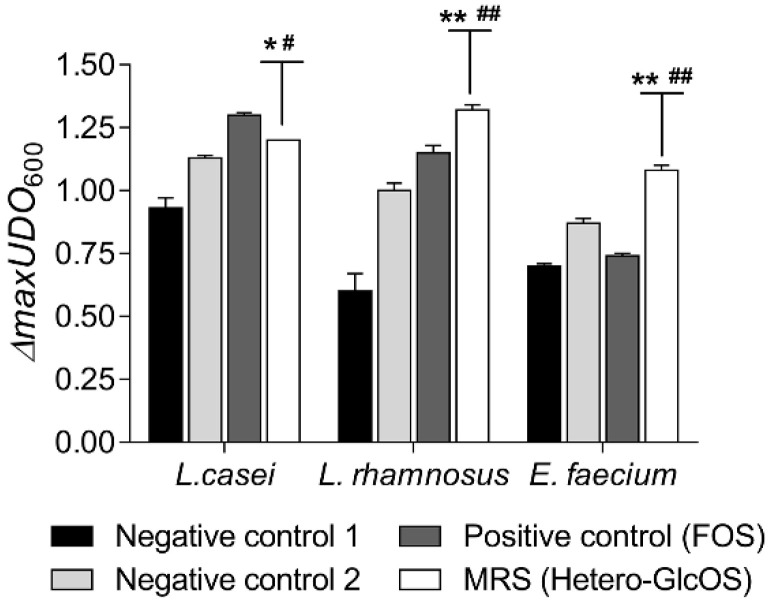
Maximal variation of turbidity (*Δ_max_*OD_600_) of lactic acid bacteria cultures. *Lactobacillus casei*, *Lactobacillus rhamnosus* and *Enterococcus faecium* were inoculated for 24 h at 37 °C in Man-Rogosa-Sharpe (MRS) medium with: no carbon source (black bars), glucose, fructose and sucrose (grey bars), the commercial FOS mixture (dark grey bars) and the mixture of hetero-GlcOS (white bars). Error bars represent the standard deviations from three independent analysis. Differences (*p* < 0.05, *p* < 0.01) with the negative and positive controls are indicated with (*) and (#), (**) and (##), respectively.

**Figure 4 ijms-23-12682-f004:**
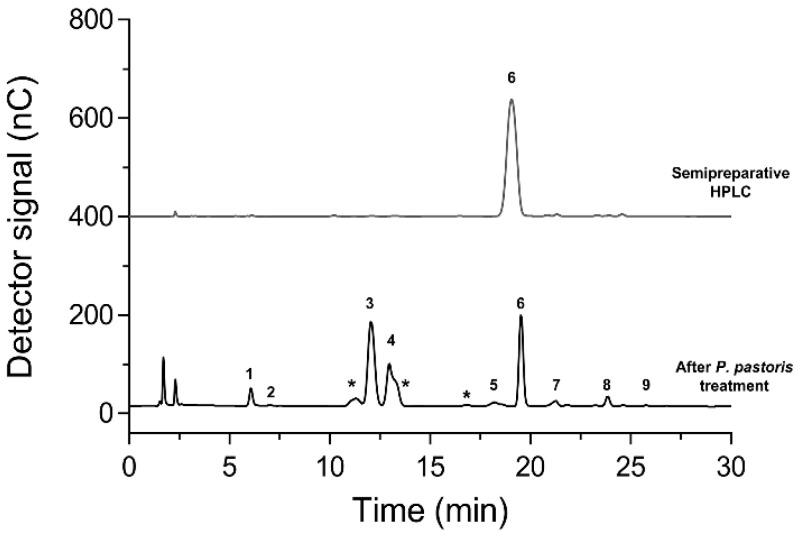
Chromatographic analysis of isomelezitose purification. HPAEC–PAD chromatograms of the hetero-GlcOS mixture treated with immobilised *K. phaffii* cells (black line) and the purified isomelezitose by semipreparative HPLC (grey line) are shown. Peaks assignation: (1) glucose, (2) fructose, (3) sucrose, (4) trehalulose, (5) melezitose, (6) isomelezitose, (7) theanderose, (8) erlose and (9) esculose. (*) unknown.

**Figure 5 ijms-23-12682-f005:**
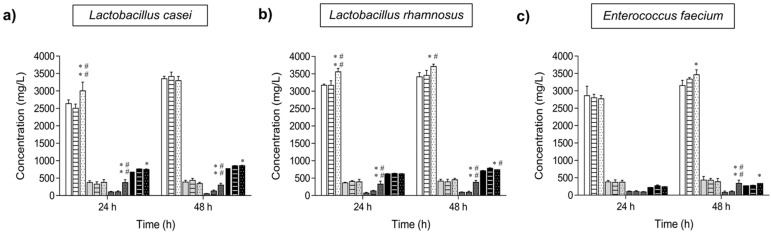
Evolution of the organic acid (SCFA and lactate) production in probiotic bacteria cultures. The concentrations of the organic acids (means ± standard deviation) at 24 and 48 h are indicated for *L. casei* (**a**), *L. rhamnosus* (**b**) and *E. faecium* (**c**) cultures in MRS media without carbon source (empty bars), with the commercial FOS mixture (lined bars) and with isomelezitose (dotted bars). References: acetate (white bars), propionate (grey bars), butyrate (dark grey bars) and lactate (blacks bars). Differences (*p* < 0.05, *p* < 0.01) with the negative and positive controls are indicated with (*) and (#), (**) and (##), respectively.

**Table 1 ijms-23-12682-t001:** Sugar composition (g/L) of transglucosylation reactions catalysed by soluble Mr-αGlu and alginate-entrapped *E. coli* cells expressing the Mr-αGlu activity.

	Glc + Fru	Suc	Tre	Isom	Rest of Hetero-GlcOS	Total Hetero-GlcOS
Entrapped *E. coli* cells	217 ± 8.9	65 ± 1.6	95 ± 3.8	75 ± 1.1	48 ± 1	218 ± 6
Soluble Mr-αGlu	97 ± 2	146 ± 16	61 ± 2	173 ± 14	23 ± 1.2	256 ± 17

Glc: glucose; Fru: fructose; Suc: sucrose; Tre: trehalulose; Isom: isomelezitose; hetero-GlcOS: hetero-glucooligosaccharides. Reaction conditions: 500 g/L sucrose in 50 mM MOPS buffer pH 7.0 and 30 °C.

**Table 2 ijms-23-12682-t002:** Growth parameters of probiotic cultures supplemented with the indicated carbon sources.

Bacteria	Grow Parameters on MRS Media					
	*Δ_max_*OD_600_^a^			*g* (h) ^b^		
	NCS	FOS	Isom	NCS	FOS	Isom
*L. casei*	0.63 ± 0.03	0.71 ± 0.03	0.76 ± 0.01 *	1.75 ± 0.01	1.34 ± 0.09	1.41 ± 0.08 *
*L. rhamnosus*	0.94 ± 0.01	1.09 ± 0.02	1.13 ± 0.02 *	1.30 ± 0.07	0.87 ± 0.04	0.80 ± 0.02 *
*E. faecium*	0.58 ± 0.03	0.79 ± 0.03	0.89 ± 0.05 *^,^^#^	1.70 ± 0.20	0.71 ± 0.14	0.87 ± 0.06 *

NCS: No Carbohydrate Source (negative control); FOS mixture Actilight (positive control); Isom (isomelezitose). Data are shown as the mean of three independent assays ± standard deviation. Tukey’s test (*p* < 0.05) was performed and significant differences with the negative control (*****) and the positive control (#) are represented. **^a^**
*Δ_max_*OD_600_ = OD_600_(24 h)-OD_600_(initial); **^b^**
*g* = generation time (h).

## Data Availability

Not applicable.

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
