# Peer review of "Isomelezitose Overproduction by Alginate-Entrapped Recombinant E. coli Cells and In Vitro Evaluation of Its Potential Prebiotic Effect"

_ijms, 2022, doi:10.3390/ijms232012682_

Round 1

Reviewer 1 Report

Dear Mrs. Férnandez-Lobato, 

I read deeply your research and I found it of relevance for the novel prebiotic field. However I feel some missing information. For instance, you did not say nothing regarding isomelezitose in the introduction part. You should explain its potential prebiotic activity (previously investigated by other authors) and also about its safety for human consumption. 

Regarding the materials and methods section I think it could be re-structured. In my opinion, section 3.4 should be divided in two parts, and section 3.3 should be in the middle of this two parts. Moreover, in section 3.8. are some indications which are not referred to any previous investigation or explained why you choose it (check lines 404, 405 and 426). It should be better explained.

Results and conclusion sections are really consistent and match the objective of this works. 

Author Response

Point-by-point response to the reviewer´s comentes:

Reviewer point 1. I read deeply your research and I found it of relevance for the novel prebiotic field. However I feel some missing information. For instance, you did not say nothing regarding isomelezitose in the introduction part. You should explain its potential prebiotic activity (previously investigated by other authors) and also about its safety for human consumption.

 Ok, corrected. New information concerning isomelezitose safety and its potential applications in the food industry was incorporated in the introduction of the new version of this manuscript (lines 98-104). In addition, at the request of evaluator number 2, new information has also been included on the enzymes that produce isomelezitose, the levels of their production and the uses of the trisaccharide (lines 82-97). Consequently, this has led to the expansion and renumbering of the cited bibliography throughout the entire manuscript.

Reviewer point 2. Regarding the materials and methods section I think it could be re-structured. In my opinion, section 3.4 should be divided in two parts, and section 3.3 should be in the middle of this two parts. Moreover, in section 3.8. are some indications which are not referred to any previous investigation or explained why you choose it (check lines 404, 405 and 426). It should be better explained.

Thanks for your comment. The sections 3.3.and 3.4 were restructured as you suggested. Additionally, in new manuscript version, section 3.9 (lines 448-449) we have indicated the research work used for the design of the prebiotic effect using in-vitro assay.

Reviewer 2 Report

Review to the manuscript entitled “Isomelezitose overproduction by alginate-entrapped recombinant E. coli cells and in vitro evaluation of its potential prebiotic effect” by Martin Garcia-Gonzalez et al. submitted to International Journal of Molecular Sciences, special issue Prebiotics and Probiotics: Healthy Biotools for Molecular Integrative and Modulation Approaches.

The manuscript of a research paper focuses on development and evaluation of an effective biocatalyst for isomelezitose production containing yeast-derived α-glucosidase overexpressed in E. coli and the recombinant cells were immobilized. Isomelezitose from the reaction mixture was further purified using yeast-based removal of monosaccharides and semipreparative HPLC. The product was further tested in cultures of prebiotic bacteria and was proved in vitro as growth- and SCFAs-enhancing compound for selected strains of gut-related probiotic bacteria.

A rare trisaccharide isomelezitose has a high potential as effective prebiotic but the limited production has created a bottleneck for its prebiotic research and potential applications. The authors have previously shown that Metschnikowia reukaufii is an efficient producer of isomelezitose and other similar oligosaccharides from sucrose that are in rather minute amounts present for example in honey.

The valorisation of sucrose into functional oligosaccharides is of a high importance for the food and chemical industry. The study has valuable content to the community of researchers focusing on enzymatic synthesis of saccharides and prebiotics. The aims and scope of the manuscript is in accordance with that of the special issue of the journal.    

The manuscript is adequately structured and composed, and clearly presented but some minor additions or corrections have to be made before the manuscript can be considered for publication. The manuscript is written in an English language on academic style.

Remarks:

1) The Introduction is generally well-organised and gives an overview of the general background and importance of the research topic. The Introduction should provide more comprehensive overview of the presence of isomelezitose in the nature and microbial enzymes that are able to convert sucrose to isomelezitose. Actually, some isomelezitose-producing enzymes from other sources have been previously biochemically characterized, the structure of α-glucosidase of yeast origin with the ability also revealed (PDB: 7P01, 7P07). This information would give valuable insight of the topic.

2) The production and purification protocol presented in the manuscript is applicable in a smaller (laboratory) scale. The use of HPLC in pilot plant-scale might not be applicable. The other options (eg membrane-based) to separate or purify oligosaccharides for food industry should be discussed.   

3) E. coli is a Gram-negative bacterium that has not under GRAS status and it has pyrogenic lipopolysaccharides (LPS). Therefore, other strategies to apply microorganisms of GRAS status has to be discussed to move forward to actual production of isomelezitose for food/pharma applications.

4) Specific comments:

Abstract, line 19. The phrase “honey oligosaccharides mixture” is partly misleading, as the substrate for the synthesis of the mixture was pure sucrose. It seems the authors want to emphasize that the product spectrum of enzymatic synthesis was similar to the saccharides in honey but in the Abstract, it seems out of the context. The word “honey” could be deleted or sucrose could be mentioned as the substrate.

Abstract, line 20 and throughout the text. The authors should specify which species they were using:  Komagataella phaffii or Komagataella pastoris. The strains of Pichia pastoris were recently divided into two species based on their genomic sequences. Based on the strain number GS115 (given in the Materials and method section, lines 304-305) the used strain should be histidine auxotrophic mutant of K. phaffii.

Lines 34-35. A considerable proportion of archaea are also present in the human gut setting.

Lines 47-49. According to the latest definition of a prebiotic, it is a substrate that is selectively utilized by host microorganisms conferring a health benefit (https://www.nature.com/articles/nrgastro.2017.75). The statement focuses on the systemic health benefit to the host and only a growth promoting agent for some probiotic strains in vitro is not a valid proof of the prebiotic effect. This should be clarified in the Introduction. The substrates for beneficial probiotic bacteria for which the health effects of the host are not yet proven should be named “candidate prebiotics” or “potential prebiotics”.     

Lines 84-85. It is not clear why the authors mention bifidogenic activity of isomelezitose when no Bifidobacteria sp. were tested in the current work.

Line 95 and throughout the manuscript. The organism names in latin should be written in italics.

Line 100. Is it necessary to call Proteus mirabilis a bacillus?

Lines 103-119. Was it confirmed that conditions showing the highest hydrolytic activity of the catalyst reflected the highest transglycosylation activity of the enzyme? Depending of the glycoside hydrolase enzyme the reaction optima for substrate hydrolysis and transglycosylation reaction.       

Fig 2. A. It is not clear which preparation the black and grey bars represent. It should be shown on the graph or in the legend.

Fig. S1., lines 161-162. The composition of an unknown product was not discussed in the Results and Discussion section. In-depth analysis of di- and trisaccharide products have been conducted on a yeast α-glucosidase products from sucrose (https://doi.org/10.3390/jof7100816), showing the range of possible products.

Lines 172-179. Was it confirmed that no manno-oligosaccharides from yeast cell walls were introduced to the mixture? Based on the information on Fig. 4, there were unknown peaks introduced to the chromatogram. This observation should be discussed in the text.

The legend of Fig. 4. There seems to be some kind of mistake in assignment of the peaks. The highest peak (6) should be isomelezitose not theanderose. The authors should review it carefully.

Lines 354-355. The speed/volume of controlled dripping should be stated.

Lines 428-429. The g values should be provided for centrifugation.  

Author Response

Point-by point response to the reviewer comments:

Remarks:

1) The Introduction is generally well-organised and gives an overview of the general background and importance of the research topic. The Introduction should provide more comprehensive overview of the presence of isomelezitose in the nature and microbial enzymes that are able to convert sucrose to isomelezitose. Actually, some isomelezitose-producing enzymes from other sources have been previously biochemically characterized, the structure of α-glucosidase of yeast origin with the ability also revealed (PDB: 7P01, 7P07). This information would give valuable insight of the topic.

We thank the reviewer for this suggestion. We find it very relevant. General information concerning isomelezitose, including its enzymatic production from sucrose, the presence of this sugar in nature and its potential application in industry was added in the new version of this manuscript (lines 88-103). Consequently, this has led to the expansion and renumbering of the cited bibliography throughout the entire manuscript.

2) The production and purification protocol presented in the manuscript is applicable in a smaller (laboratory) scale. The use of HPLC in pilot plant-scale might not be applicable. The other options (eg membrane-based) to separate or purify oligosaccharides for food industry should be discussed.  

Ok, this issue has been discussed in the new version of the manuscript. Nanofiltration as a cost-effective and easy to scale-up methodology for oligosaccharide purification was commented, and different examples of bioactive oligosaccharides purified using this strategy were also described in section 2.4 (lines 253-262).

3) E. coli is a Gram-negative bacterium that has not under GRAS status and it has pyrogenic lipopolysaccharides (LPS). Therefore, other strategies to apply microorganisms of GRAS status has to be discussed to move forward to actual production of isomelezitose for food/pharma applications.

Thanks again for the suggestion. The overexpression of the glycoside hydrolase Mr-αGlu in GRAS organisms such as S. cerevisiae or K. phaffii, and the further use of the transformed cells of these organisms as immobilized biocatalysts was also discussed in the section 2.1 (lines 154-158).

4) Specific comments:

Abstract, line 19. The phrase “honey oligosaccharides mixture” is partly misleading, as the substrate for the synthesis of the mixture was pure sucrose. It seems the authors want to emphasize that the product spectrum of enzymatic synthesis was similar to the saccharides in honey but in the Abstract, it seems out of the context. The word “honey” could be deleted, or sucrose could be mentioned as the substrate.

Ok, corrected. The sentence was rewritten, and the word “honey” deleted (lines 18-19).

Abstract, line 20 and throughout the text. The authors should specify which species they were using:  Komagataella phaffii or Komagataella pastoris. The strains of Pichia pastoris were recently divided into two species based on their genomic sequences. Based on the strain number GS115 (given in the Materials and method section, lines 304-305) the used strain should be histidine auxotrophic mutant of K. phaffii.

Ok, thanks for the comment.  In line with the current taxonomy classification the yeast Pichia pastoris was referenced as Komagataella phaffii throughout the entire manuscript.

Lines 34-35. A considerable proportion of archaea are also present in the human gut setting.

Ok, corrected. This information was included and consequently the bibliography was updated (lines 35-37).

Lines 47-49. According to the latest definition of a prebiotic, it is a substrate that is selectively utilized by host microorganisms conferring a health benefit (https://www.nature.com/articles/nrgastro.2017.75). The statement focuses on the systemic health benefit to the host and only a growth promoting agent for some probiotic strains in vitro is not a valid proof of the prebiotic effect. This should be clarified in the Introduction. The substrates for beneficial probiotic bacteria for which the health effects of the host are not yet proven should be named “candidate prebiotics” or “potential prebiotics”.

Ok, thanks for this very fair point. These aspects were corrected in the new manuscript version (lines 49-53).

Lines 84-85. It is not clear why the authors mention bifidogenic activity of isomelezitose when no Bifidobacteria sp. were tested in the current work. Ok, corrected. the term “bifidogenic” was eliminated or substituted by “(potential) prebiotic” throughout all the manuscript text, as in lines 68, 104 and 213 of the new manuscript version.

Line 95 and throughout the manuscript. The organism names in latin should be written in italics.

Ok, corrected. All the organisms’ names were checked and written in italics.

Line 100. Is it necessary to call Proteus mirabilis a bacillus?

OK, corrected.

Lines 103-119. Was it confirmed that conditions showing the highest hydrolytic activity of the catalyst reflected the highest transglycosylation activity of the enzyme? Depending of the glycoside hydrolase enzyme the reaction optima for substrate hydrolysis and transglycosylation reaction. We agree with the comment made by the reviewer, but this aspect of the enzyme has not yet been addressed. In previous works, we found that the transglucosylation activity of the free Mr-αGlu was mainly dependent on both, substrate concentration and temperature. But concerning this part of the new work we only analysed the hydrolytic activity of the enzyme expressed by alginate-entrapped E.coli cells using the optimal temperature determined for the soluble enzyme. In this case, the amount of entrapped E. coli cells was evaluated in order to obtained the biocatalyst with the highest apparent hydrolytic activity (a measure of the active enzyme in the immobilized biocatalyst) in order to scale the transfer reactions.  

Fig 2. A. It is not clear which preparation the black and grey bars represent. It should be shown on the graph or in the legend.

Ok, corrected. The legend of the Figure 2 has been modified in order to clarify what represent each bar (lines 168-169).

Fig. S1, lines 161-162. The composition of an unknown product was not discussed in the Results and Discussion section. In-depth analysis of di- and trisaccharide products have been conducted on a yeast α-glucosidase products from sucrose (https://doi.org/10.3390/jof7100816), showing the range of possible products.

Ok, thanks for the observation. Considering the information from the experiments carried out by Ernits et al. (2021), the possible identity of the unknown peak detected in our chromatographic analysis was indicated and this work referenced in our manuscript text (lines 185-190).

Lines 172-179. Was it confirmed that no manno-oligosaccharides from yeast cell walls were introduced to the mixture? Based on the information on Fig. 4, there were unknown peaks introduced to the chromatogram. This observation should be discussed in the text.

Ok, thanks for the comment. This observation has already been discussed in the new manuscript version, after information concerning Figure 4 (lines 248-252).

The legend of Fig. 4. There seems to be some kind of mistake in assignment of the peaks. The highest peak (6) should be isomelezitose not theanderose. The authors should review it carefully. Thank you very much for the annotation, clearly there was an error in the assignation of peaks in the Figure 4 legend that has been corrected in the new manuscript version.

Lines 354-355. The speed/volume of controlled dripping should be stated. Ok, the peristaltic pump used for the production of the alginate beads and the flow fixed for it was now included in the new manuscript version (line 401).

Lines 428-429. The g values should be provided for centrifugation.  Ok, corrected. The corresponding g value for centrifugation was indicated in lines 476-477).

Round 2

Reviewer 2 Report

Comment to the revised manuscript by Martin Garcia-Gonzalez: I sincerely thank the authors for their effort in careful revision of the manuscript. The manuscript reach, presentation of the content and discussion have been considerably improved during the revision. The issues and comments that were pointed out have been fully addressed by the authors.

There are only some very minor remarks:

Line 53. Typographical error in “in vivo”.

Line 258. Typographical error in “xylo-oligosaccharides”.